# HRFormer: High-Resolution Transformer for Dense Prediction

**Yuhui Yuan**[124]    **Rao Fu**[1]    **Lang Huang**[3]    **Weihong Lin**[4]    **Chao Zhang**[3]    **Xilin Chen**[12]
**Jingdong Wang**[5]*

[1]University of Chinese Academy of Sciences    [2]Institute of Computing Technology, CAS
[3]Peking University    [4]Microsoft Research Asia    [5]Baidu

## Abstract

We present a High-Resolution Transformer (HRFormer) that learns high-resolution representations for dense prediction tasks, in contrast to the original Vision Transformer that produces low-resolution representations and has high memory and computational cost. We take advantage of the multi-resolution parallel design introduced in high-resolution convolutional networks (HRNet [46]), along with local-window self-attention that performs self-attention over small non-overlapping image windows [21], for improving the memory and computation efficiency. In addition, we introduce a convolution into the FFN to exchange information across the disconnected image windows. We demonstrate the effectiveness of the High-Resolution Transformer on both human pose estimation and semantic segmentation tasks, e.g., HRFormer outperforms Swin transformer [27] by 1.3 AP on COCO pose estimation with 50% fewer parameters and 30% fewer FLOPs. Code is available at: https://github.com/HRNet/HRFormer.

## 1 Introduction

Vision Transformer (ViT) [13] shows promising performance on ImageNet classification tasks. Many follow-up works boost the classification accuracy through knowledge distillation [42], adopting deeper architecture [43], directly introducing convolution operations [16, 48], redesigning input image tokens [54], and etc. Besides, some studies attempt to extend the transformer to address broader vision tasks such as object detection [4], semantic segmentation [63, 37], pose estimation [51, 23], video understanding [61, 2, 30], and so on. This work focuses on the transformer for dense prediction tasks, including pose estimation and semantic segmentation.

Vision Transformer splits an image into a sequence of image patches of size $16 \times 16$, and extracts the feature representation of each image patch. Thus, the output representations of Vision Transformer lose the fine-grained spatial details that are essential for accurate dense predictions. The Vision Transformer only outputs a single-scale feature representation, and thus lacks the capability to handle multi-scale variation. To mitigate the loss of feature granularity and model the multi-scale variation, we present High-Resolution Transformer (HRFormer) that contains richer spatial information and constructs multi-resolution representations for dense predictions.

The High-Resolution Transformer is built by following the multi-resolution parallel design that is adopted in HRNet [46]. First, HRFormer adopts convolution in both the stem and the first stage as several concurrent studies [11, 50] also suggest that convolution performs better in the early stages. Second, HRFormer maintains a high-resolution stream through the entire process with parallel medium- and low-resolution streams helping boost high-resolution representations. With feature maps of different resolutions, thus HRFormer is capable to model the multi-scale variation. Third,

---

*Corresponding author.

35th Conference on Neural Information Processing Systems (NeurIPS 2021).

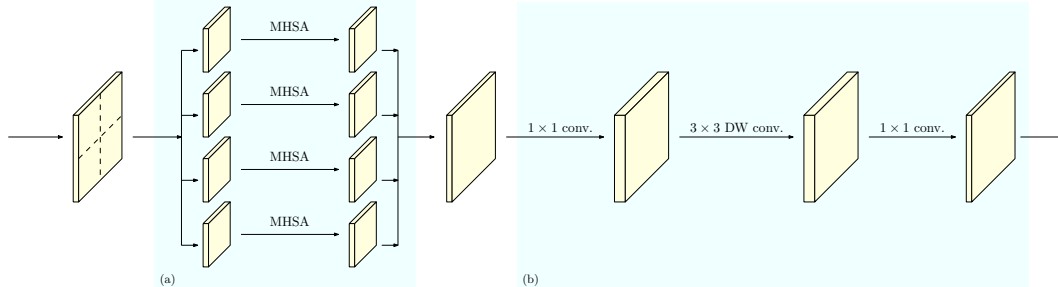

Figure 1: **Illustrating the HRFormer block.** The HRFormer block is composed of (a) local-window self-attentionm and (b) feed-forward network (FFN) with depth-wise convolution. The local-window self-attention scheme is inspired by the interlaced sparse self-attention [56, 21].

HRFormer mixes the short-range and long-range attention via exchanging multi-resolution feature information with the multi-scale fusion module.

At each resolution, the local-window self-attention mechanism is adopted to reduce the memory and computation complexity. We partition the representation maps into a set of non-overlapping small image windows and perform self-attention in each image window separately. This reduces the memory and computation complexity from quadratic to linear with respect to spatial size. We further introduce a $3 \times 3$ depth-wise convolution into the feed-forward network (FFN) that follows the local-window self-attention, to exchange information between the image windows which are disconnected in the local-window self-attention process. This helps to expand the receptive field and is essential for dense prediction tasks. Figure 1 shows the details of an HRFormer block.

We conduct experiments on image classification, pose estimation, and semantic segmentation tasks, and achieve competitive performance on various benchmarks. For example, HRFormer-B gains $+1.0\%$ top-1 accuracy on ImageNet classification over DeiT-B [42] with $40\%$ fewer parameters and $20\%$ fewer FLOPs. HRFormer-B gains $0.9\%$ AP over HRNet-W48 [41] on COCO `val` set with with $32\%$ fewer parameters and $19\%$ fewer FLOPs. HRFormer-B + OCR gains $+1.2\%$ and $+2.0\%$ mIoU over HRNet-W48 + OCR [55] with $25\%$ fewer parameters and slightly more FLOPs on PASCAL-Context `test` and COCO-Stuff `test`, respectively.

## 2 Related work

**Vision Transformers.** With the success of Vision Transformer (ViT) [13] and the data-efficient image transformer (DeiT) [42], various techniques are proposed to improve the ImageNet classification accuracy of Vision Transformer [12, 43, 48, 16, 54, 17, 5, 27, 22, 40]. Among the very recent advancements, the community has verified several effective improvements such as multi-scale feature hierarchies and incorporating convolutions.

For example, the concurrent works MViT [14], PVT [47], and Swin [27] introduce the multi-scale feature hierarchies into transformer following the spatial configuration of a typical convolutional architecture such as ResNet-50. Different from them, our HRFormer incorporates the multi-scale feature hierarchies through exploiting the multi-resolution parallel design inspired by HRNet. CvT [48], CeiT [53], and LocalViT [25] propose to enhance the locality of transformer via inserting depth-wise convolutions into either the self-attention or the FFN. The purpose of the inserted convolution within our HRFormer is different, apart from enhancing the locality, it also ensures information exchange across the non-overlapping windows.

Several previous studies [36, 19] have proposed similar local self-attention schemes for image classification. They construct the overlapped local windows following the strided convolution, resulting in heavy computation cost. Similar to [21, 44, 27], we propose to apply the local-window self-attention scheme to divide the input feature map into non-overlapping windows. Then we apply the self-attention within each window independently so as to improve the efficiency significantly.

There are several concurrently-developed works [63, 37] use the Vision Transformer to address the dense predict tasks such as semantic segmentation. They have shown that increasing the spatial resolution of the representations output by the Vision Transformer is important for semantic segmen-

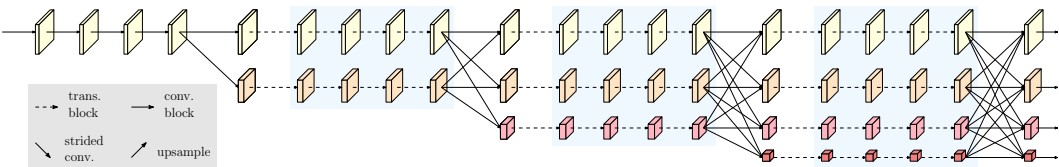

Figure 2: **Illustrating the High-Resolution Transformer architecture.** The multi-resolution parallel transformer modules are marked with light blue color areas. Each module consists of multiple successive multi-resolution parallel transformer blocks. The first stage is constructed with convolution block and the remained three stages are constructed with transformer block.

tation. Our HRFormer provides a different path to address the low-resolution problem of the Vision Transformer via exploiting the multi-resolution parallel transformer scheme.

**High-Resolution CNN for Dense Prediction.** The high-resolution convolutional schemes have achieved great success on both pose estimation and semantic segmentation tasks. In the development of high-resolution convolutional neural networks, the community has developed three main paths including: (i) applying dilated convolutions to remove some down-sample layers [6, 52], (ii) recovering high-resolution representations from low-resolution representations with decoders [38, 1, 31, 32], and (iii) maintaining high-resolution representations throughout the network [46, 15, 39, 64, 45, 59, 20]. Our HRFormer belongs to the third path, and retains the advantages of both vision transformer and HRNet [46].

## 3 High-Resolution Transformer

**Multi-resolution parallel transformer.** We follow the HRNet [46] design and start from a high-resolution convolution stem as the first stage, gradually adding high-to-low resolution streams one by one as new stages. The multi-resolution streams are connected in parallel. The main body consists of a sequence of stages. In each stage, the feature representation of each resolution stream is updated with multiple transformer blocks independently and the information across resolutions is exchanged repeatedly with the convolutional multi-scale fusion modules.

Figure 2 illustrates the overall HRFormer architecture. The design of convolutional multi-scale fusion modules exactly follows HRNet. We illustrate the details of the transformer block in the following discussion and more details are presented in Figure 1.

**Local-window self-attention.** We divide the feature maps $\mathbf{X} \in \mathbb{R}^{N \times D}$ into a set of non-overlapping small windows: $\mathbf{X} \to \{\mathbf{X}_1, \mathbf{X}_2, \cdots, \mathbf{X}_P\}$, where each window is of size $K \times K$. We perform multi-head self-attention (MHSA) within each window independently. The formulation of multi-head self-attention on the $p$-th window is given as:

$$\text{MultiHead}(\mathbf{X}_p) = \text{Concat}[\text{head}(\mathbf{X}_p)_1, \cdots, \text{head}(\mathbf{X}_p)_H] \in \mathbb{R}^{K^2 \times D}, \tag{1}$$

$$\text{head}(\mathbf{X}_p)_h = \text{Softmax}\left[\frac{(\mathbf{X}_p \mathbf{W}_q^h)(\mathbf{X}_p \mathbf{W}_k^h)^T}{\sqrt{D/H}}\right] \mathbf{X}_p \mathbf{W}_v^h \in \mathbb{R}^{K^2 \times \frac{D}{H}}, \tag{2}$$

$$\widehat{\mathbf{X}}_p = \mathbf{X}_p + \text{MultiHead}(\mathbf{X}_p)\mathbf{W}_o \in \mathbb{R}^{K^2 \times \frac{D}{H}}, \tag{3}$$

where $\mathbf{W}_o \in \mathbb{R}^{D \times D}$, $\mathbf{W}_q^h \in \mathbb{R}^{\frac{D}{H} \times D}$, $\mathbf{W}_k^h \in \mathbb{R}^{\frac{D}{H} \times D}$, and $\mathbf{W}_v^h \in \mathbb{R}^{\frac{D}{H} \times D}$ for $h \in \{1, \cdots, H\}$. $H$ represents the number of heads, $D$ represents the number of channels, $N$ represents the input resolutions, and $\widehat{\mathbf{X}}_p$ represents the output representation of MHSA. We also apply the relative position embedding scheme introduced in the T5 model [35] to incorporate the relative position information into the local-window self-attention.

With MHSA aggregates information within each window, we merge them to compute the output $\mathbf{X}^{\text{MHSA}}$:

$$\{\widehat{\mathbf{X}}_1, \widehat{\mathbf{X}}_2, \cdots, \widehat{\mathbf{X}}_P\} \xrightarrow{\text{Merge}} \mathbf{X}^{\text{MHSA}}. \tag{4}$$

The left part of Figure 1 illustrates how local-window self-attention updates the 2D input representations, where the multi-head self-attention operates within each window independently.

Table 1: **The architecture configuration of HRFormer.** LSA: local-window self-attention, FFN-DW: feed-forward network with a $3 \times 3$ depth-wise convolution, $(M_1, M_2, M_3, M_4)$: the number of modules, $(B_1, B_2, B_3, B_4)$: the number of blocks, $(W_1, W_2, W_3, W_4)$: the size of windows, $(H_1, H_2, H_3, H_4)$: the number of heads, $(R_1, R_2, R_3, R_4)$: the MLP expansion ratios.

| Res. | Stage 1 | Stage 2 | Stage 3 | Stage 4 |
|---|---|---|---|---|
| $4\times$ | $\begin{bmatrix} 1 \times 1, 64 \\ 3 \times 3, 64 \\ 1 \times 1, 256 \end{bmatrix} \times B_1 \times M_1$ | $\begin{bmatrix} \text{LSA}, W_1, H_1 \\ \text{FFN-DW}, R_1 \end{bmatrix} \times B_2 \times M_2$ | $\begin{bmatrix} \text{LSA}, W_1, H_1 \\ \text{FFN-DW}, R_1 \end{bmatrix} \times B_3 \times M_3$ | $\begin{bmatrix} \text{LSA}, W_1, H_1 \\ \text{FFN-DW}, R_1 \end{bmatrix} \times B_4 \times M_4$ |
| $8\times$ | | $\begin{bmatrix} \text{LSA}, W_2, H_2 \\ \text{FFN-DW}, R_2 \end{bmatrix} \times B_2 \times M_2$ | $\begin{bmatrix} \text{LSA}, W_2, H_2 \\ \text{FFN-DW}, R_2 \end{bmatrix} \times B_3 \times M_3$ | $\begin{bmatrix} \text{LSA}, W_2, H_2 \\ \text{FFN-DW}, R_2 \end{bmatrix} \times B_4 \times M_4$ |
| $16\times$ | | | $\begin{bmatrix} \text{LSA}, W_3, H_3 \\ \text{FFN-DW}, R_3 \end{bmatrix} \times B_3 \times M_3$ | $\begin{bmatrix} \text{LSA}, W_3, H_3 \\ \text{FFN-DW}, R_3 \end{bmatrix} \times B_4 \times M_4$ |
| $32\times$ | | | | $\begin{bmatrix} \text{LSA}, W_4, H_4 \\ \text{FFN-DW}, R_4 \end{bmatrix} \times B_4 \times M_4$ |

Table 2: **HRFormer instances.** HRFormer-T, HRFormer-S, and HRFormer-B represents tiny, small, and base HRFormer model, respectively.

| Model | #modules $(M_1, M_2, M_3, M_4)$ | #blocks $(B_1, B_2, B_3, B_4)$ | #channels $(C_1, C_2, C_3, C_4)$ | #heads $(H_1, H_2, H_3, H_4)$ |
|---|---|---|---|---|
| HRFormer-T | $(1, 1, 3, 2)$ | $(2, 2, 2, 2)$ | $(18, 36, 72, 144)$ | $(1, 2, 4, 8)$ |
| HRFormer-S | $(1, 1, 4, 2)$ | $(2, 2, 2, 2)$ | $(32, 64, 128, 256)$ | $(1, 2, 4, 8)$ |
| HRFormer-B | $(1, 1, 4, 2)$ | $(2, 2, 2, 2)$ | $(78, 156, 312, 624)$ | $(2, 4, 8, 16)$ |

**FFN with depth-wise convolution.** Local-window self-attention performs self-attention over the non-overlapping windows separately. There is no information exchange across the windows. To handle this issue, we add a $3 \times 3$ depth-wise convolution between the two point-wise MLPs that form the FFN in Vision transformer: $\mathrm{MLP}(\mathrm{DW\text{-}Conv.}(\mathrm{MLP}()))$. The right part of Figure 1 shows an example of how FFN with $3 \times 3$ depth-wise convolution updates the 2D input representations.

**Representation head designs.** As shown in Figure 2, the output of HRFormer consists of four feature maps of different resolutions. We illustrate the details of the representation head designs for different tasks as following: (i) ImageNet classification, we send the four-resolution feature maps into a bottleneck and the output channels are changed to 128, 256, 512, and 1024 respectively. Then, we apply the strided convolutions to fuse them and output a feature map of the lowest resolution with 2048 channels. Last, we apply a global average pooling operation followed by the final classifier. (ii) pose estimation, we only apply the regression head over the highest resolution feature map. (iii) semantic segmentation, we apply the semantic segmentation head over the concatenated representations, which are computed by first upsampling all the low-resolution representations to the highest resolution and then concatenate them together.

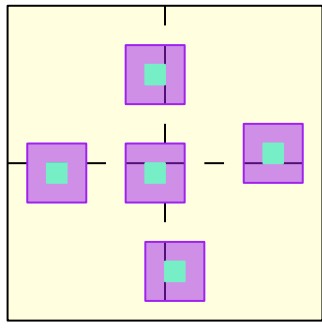

Figure 3: Illustrating that FFN with $3 \times 3$ depth-wise convolution connects the non-overlapping windows.

**Instantiation.** We illustrate the overall architecture configuration of HRFormer in Table 1. We use $(M_1, M_2, M_3, M_4)$ and $(B_1, B_2, B_3, B_4)$ to represent the number of modules and the number of blocks of {state1, stage2, stage3, stage4}, respectively. We use $(C_1, C_2, C_3, C_4)$, $(H_1, H_2, H_3, H_4)$ and $(R_1, R_2, R_3, R_4)$ to represent the number of channels, the number of heads and the MLP expansion ratios in transformer block associated with different resolutions. We keep the first stage unchanged following the original HRNet and use the bottleneck as the basic building block. We apply the transformer blocks in the other stages and each transformer block consists of a local-window self-attention followed by an FFN with $3 \times 3$

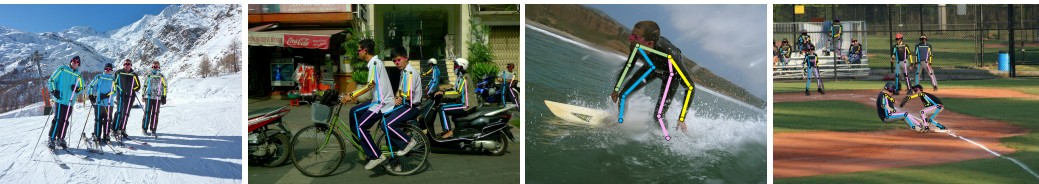

Figure 4: Example results of HRFormer-B on COCO pose estimation `val`: containing occlusion, multiple persons, viewpoint and appearance change.

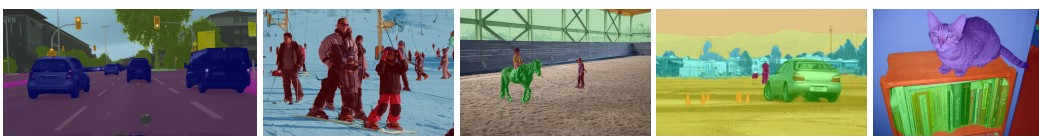

Figure 5: Example results of HRFormer-B + OCR on Cityscapes `val` (left one), COCO-Stuff `test` (middle two), and PASCAL-Context `test` (right two).

depth-wise convolution. We have not included the convolutional multi-scale fusion modules in Table 1 for simplicity. In our implementation, we set the size of the windows on four resolution streams as $(7, 7, 7, 7)$ by default. Table 2 illustrates the configuration details of three different HRFormer instances with increasing complexities, where the MLP expansion ratios $(R_1, R_2, R_3, R_4)$ are set as $(4, 4, 4, 4)$ for all models and are not shown.

**Analysis.** The benefits of $3 \times 3$ depth-wise convolution are twofold: one is enhancing the locality and the other one is enabling the interactions across windows. We illustrate how the FFN with depth-wise convolution is capable to expand the interactions beyond the non-overlapping local windows and model the relations between them in Figure 3. Therefore, based on the combination of the local-window self-attention and the FFN with $3 \times 3$ depth-wise convolution, we can build the HRFormer block that improves the memory and computation efficiency significantly.

## 4 Experiments

### 4.1 Human Pose Estimation

**Training setting.** We study the performance of HRFormer on the COCO [26] human pose estimation benchmark, which contains more than 200K images and 250K person instances labeled with 17 keypoints. We train our model on COCO `train` 2017 dataset, including 57K images and 150K person instances. We evaluate our approach on the `val` 2017 set and `test-dev` 2017, containing 5K images and 20K images, respectively.

We follow most of the default training and evaluation settings of `mmpose` [8][2], and change the optimizer from Adam to AdamW. For the training batch size, we choose 256 for HRFormer-T and HRFormer-S and 128 for HRFormer-B due to limited GPU memory. Each HRFormer experiment on COCO pose estimation task takes $8 \times$ 32G-V100 GPUs.

**Results.** Table 3 reports the comparisons on COCO `val` set. We compare HRFormer to the representative convolutional method such as HRNet [41] and several recent transformer methods, including PRTR [23], TransPose-H-A6 [51], and TokenPose-L/D24 [24]. HRFormer-B gains $0.9\%$ with $32\%$ fewer parameters and $19\%$ fewer FLOPs when compared to HRNet-W48 with an input size of $384 \times 288$. Therefore, our HRFormer-B already achieves $77.2\%$ w/o using any advanced techniques such as UDP [20] and DARK[59]. We believe that our HRFormer-B could achieve better results by exploiting either UDP or DARK scheme. We also report the comparisons on COCO `test-dev` set in Table 4. Our HRFormer-B outperforms HRNet-W48 by around $0.7\%$ with fewer parameters and FLOPs. Figure 4 shows some example results of human pose estimation on COCO `val` set.

---

[2]https://github.com/open-mmlab/mmpose, Apache License 2.0

Table 3: **Comparison on the COCO pose estimation** `val` **set.** The number of parameters and FLOPs for the pose estimation network are measured w/o considering neither human detection nor keypoint grouping. All results are based on ImageNet pretraining. $-$ means the numbers are not provided in the original paper.

| Method | input size | #param. | FLOPs | AP | $AP^{50}$ | $AP^{75}$ | $AP^{M}$ | $AP^{L}$ | AR |
|---|---|---|---|---|---|---|---|---|---|
| HRNet-W32 [41] | $256 \times 192$ | 28.5M | 7.1G | 74.4 | 90.5 | 81.9 | 70.8 | 81.0 | 78.9 |
| HRNet-W32 [41] | $384 \times 288$ | 28.5M | 16.0G | 75.8 | 90.6 | 82.7 | 71.9 | 82.8 | 81.0 |
| HRNet-W48 [41] | $256 \times 192$ | 63.6M | 14.6G | 75.1 | 90.6 | 82.2 | 71.5 | 81.8 | 80.4 |
| HRNet-W48 [41] | $384 \times 288$ | 63.6M | 32.9G | 76.3 | 90.8 | 82.9 | 72.3 | 83.4 | 81.2 |
| PRTR [23] | $512 \times 384$ | 57.2M | 37.8G | 73.3 | 89.2 | 79.9 | 69.0 | 80.9 | 80.2 |
| TransPose-H-A6 [51] | $256 \times 192$ | 17.5M | 21.8G | 75.8 | $-$ | $-$ | $-$ | $-$ | 80.8 |
| TokenPose-L/D24 [24] | $256 \times 192$ | 27.5M | 11.0G | 75.8 | 90.3 | 82.5 | 72.3 | 82.7 | 80.9 |
| HRFormer-T | $256 \times 192$ | 2.5M | 1.3G | 70.9 | 89.0 | 78.4 | 67.2 | 77.8 | 76.6 |
| HRFormer-T | $384 \times 288$ | 2.5M | 1.8G | 72.4 | 89.3 | 79.0 | 68.2 | 79.7 | 77.9 |
| HRFormer-S | $256 \times 192$ | 7.8M | 2.8G | 74.0 | 90.2 | 81.2 | 70.4 | 80.7 | 79.4 |
| HRFormer-S | $384 \times 288$ | 7.8M | 6.2G | 75.6 | 90.3 | 82.2 | 71.6 | 82.5 | 80.7 |
| HRFormer-B | $256 \times 192$ | 43.2M | 12.2G | 75.6 | 90.8 | 82.8 | 71.7 | 82.6 | 80.8 |
| HRFormer-B | $384 \times 288$ | 43.2M | 26.8G | 77.2 | 91.0 | 83.6 | 73.2 | 84.2 | 82.0 |

Table 4: **Comparison on the COCO pose estimation** `test-dev` **set.** The number of parameters and FLOPs for the pose estimation network are measured w/o considering neither human detection nor keypoint grouping. All results are based on ImageNet pretraining.

| Method | input size | #param. | FLOPs | AP | $AP^{50}$ | $AP^{75}$ | $AP^{M}$ | $AP^{L}$ | AR |
|---|---|---|---|---|---|---|---|---|---|
| HRNet-W32 [41] | $384 \times 288$ | 28.5M | 16.0G | 74.9 | 92.5 | 82.8 | 71.3 | 80.9 | 80.1 |
| HRNet-W48 [41] | $384 \times 288$ | 63.6M | 32.9G | 75.5 | 92.5 | 83.3 | 71.9 | 81.5 | 80.5 |
| PRTR [23] | $512 \times 384$ | 57.2M | 37.8G | 72.1 | 90.4 | 79.6 | 68.1 | 79.0 | 79.4 |
| TransPose-H-A6 [51] | $256 \times 192$ | 17.5M | 21.8G | 75.0 | 92.2 | 82.3 | 71.3 | 81.1 | $-$ |
| TokenPose-L/D24 [24] | $384 \times 288$ | 29.8M | 22.1G | 75.9 | 92.3 | 83.4 | 72.2 | 82.1 | 80.8 |
| HRFormer-S | $384 \times 288$ | 7.8M | 6.2G | 74.5 | 92.3 | 82.1 | 70.7 | 80.6 | 79.8 |
| HRFormer-B | $384 \times 288$ | 43.2M | 26.8G | 76.2 | 92.7 | 83.8 | 72.5 | 82.3 | 81.2 |

## 4.2 Semantic Segmentation

**Cityscapes**. The Cityscapes dataset [9] is for urban scene understanding. There are a total of 30 classes and only 19 classes are used for parsing evaluation. The dataset contains 5K high-quality pixel-level finely annotated images and 20K coarsely annotated images. The finely annotated 5K images are divided into $2,975$ `train` images, $500$ `val` images and $1,525$ `test` images. We set the initial learning rate as $0.0001$, weight decay as $0.01$, crop size as $1024 \times 512$, batch size as 8, and training iterations as 80K by default. Each HRFormer + OCR experiment on Cityscapes takes $8\times$ 32G-V100 GPUs.

Table 5 reports the results on Cityscapes `val`. We choose to use HRFormer + OCR as our semantic segmentation architecture. We compare our method with several well-known Vision Transformer based methods [63, 37] and CNN based methods [6, 62, 55]. Specifically, SETR-PUP and SETR-MLA use the ViT-Large [13] as the backbone. DPT-Hybrid uses the ViT-Hybrid [13] that consists of a ResNet-50 followed by 12 transformer layers. Both ViT-Large and ViT-Hybrid are initialized with the weights pre-trained on ImageNet-21K, where both of them achieve around $85.1\%$ top1 accuracy on ImageNet. DeepLabv3 [6] and PSPNet [62] are based on dilated ResNet-101 with output stride 8. According to the fourth column of Table 5, HRFormer + OCR achieves competitive performance overall. For example, HRFormer-B + OCR achieves comparable performance with SETR-PUP while saving $70\%$ parameters and $50\%$ FLOPs.

Table 5: **Comparison with the recent SOTA on semantic segmentation tasks.** We report the mIoUs on Cityscapes `val`, PASCAL-Context `test`, COCO-Stuff `test`, and ADE20K `val`. The number of parameters and FLOPs are measured on the image size of $1024 \times 1024$, and the output label map size of $19 \times 1024 \times 1024$. All results are evaluated with multi-scale testing. ‡: the results are obtained with extra pre-training on ADE20K.

| Method | #params. | FLOPs | Cityscapes | PASCAL-Context | COCO-Stuff | ADE20K |
|---|---|---|---|---|---|---|
| *Transformer backbone* | | | | | | |
| SETR-PUP [63] | 317.8M | 2326.7G | 82.2 | 55.3 | − | 50.1 |
| SETR-MLA [63] | 309.5M | 2138.6G | − | 55.8 | − | 50.3 |
| Swin-S + UperNet [27] | 81.16M | 1036.50G | − | − | − | 49.5 |
| Swin-B + UperNet [27] | 121.18M | 1187.90G | − | − | − | 49.7 |
| PVT-Large + Semantic FPN [47] | 65.1M | −G | − | − | − | 43.5 |
| *CNN backbone* | | | | | | |
| Deeplabv3 [7] | 87.1M | 1394.0G | 80.7 | 54.1 | − | − |
| PSPNet [62] | 68.0M | 1028.8G | 80.0 | 54.0 | 43.3 | − |
| HRNet-W48 + OCR [55] | 74.5M | 924.7G | − | 56.2 | 40.5 | 45.7 |
| *CNN+Transformer backbone* | | | | | | |
| DPT-Hybrid [37] | 124.0M | 1231.5G | − | 60.5‡ | − | 49.0 |
| HRFormer-B + OCR | 56.2M | 1119.9G | 82.6 | 58.5 | 43.3 | 50.0 |
| HRFormer-B + OCR + SegFix [57] | 56.2M | 1119.9G | 83.2 | − | − | − |

**PASCAL-Context**. The PASCAL-Context dataset [29] is a challenging scene parsing dataset that contains 59 semantic classes and 1 background class. The `train` set and `test` set consist of $4,998$ and $5,105$ images respectively. We set the initial learning rate as $0.0001$, weight decay as $0.01$, crop size as $520 \times 520$, batch size as 16, and training iterations as 60K by default. We report the comparisons on the fifth column of Table 5. Accordingly, HRFormer-B + OCR gains $1.1\%$, $1.5\%$ over HRNet-W48 + OCR, SETR-MLA with fewer parameters and FLOPs, respectively. Notably, DPT-Hybrid achieves the best performance through extra pre-training the models on ADE20K in advance. Each HRFormer + OCR experiment on PASCAL-Context takes $8 \times$ 32G-V100 GPUs.

**COCO-Stuff**. The COCO-Stuff dataset [3] is a challenging scene parsing dataset that contains 171 semantic classes. The `train` set and `test` set consist of 9K and 1K images respectively. We set the initial learning rate as $0.0001$, weight decay as $0.01$, crop size as $520 \times 520$, batch size as 16, and training iterations as 60K by default. We report the comparisons on the last column of Table 5 and HRFormer-B + OCR outperforms the previous best-performing HRNet-W48 + OCR by nearly $2\%$. Each HRFormer + OCR experiment on COCO-Stuff takes $8 \times$ 32G-V100 GPUs. Figure 5 shows some example results on Cityscapes, PASCAL-Context, and COCO-Stuff.

### 4.3 ImageNet Classification

**Training setting.** We conduct the comparisons on ImageNet-1K, which consists of 1.28M `train` images and 50K `val` images with 1000 classes. We train all models with batch size 1024 for 300 epochs with AdamW [28] optimizer, cosine decay learning rate schedule, weight decay as $0.05$, and a bag of augmentation policies, including rand augmentation [10], mixup [60], cutmix [58], and so on. HRFormer-T and HRFormer-S require $8 \times$ 32G-V100 GPUs and HRFormer-B requires $32 \times$ 32G-V100 GPUs.

**Results.** We compare HRFormer to some representative CNN methods and vision transformer methods in Table 6, where all methods are trained on ImageNet-1K only. The results of ViT-Large with larger dataset such as ImageNet-21K not included for fairness. According to Table 6, HRFormer achieves competitive performance. For example, HRFormer-B gains $1.0\%$ over DeiT-B while saving nearly $40\%$ parameters and $20\%$ FLOPs.

Table 6: **Comparisons on ImageNet-1K val.**

| Method | image size | #param. | FLOPs | Top-1 acc. |
|---|---|---|---|---|
| ResNet-18 [18] | $224 \times 224$ | 11M | 1.8G | 69.8 |
| ResNet-50 [18] | $224 \times 224$ | 26M | 4.1G | 78.5 |
| ResNet-101 [18] | $224 \times 224$ | 45M | 7.9G | 79.8 |
| HRNet-W18 [46] | $224 \times 224$ | 21.3M | 4.0G | 76.8 |
| HRNet-W32 [46] | $224 \times 224$ | 41.2M | 8.3G | 78.5 |
| HRNet-W48 [46] | $224 \times 224$ | 77.5M | 16.1G | 79.3 |
| RegNetY-4G [34] | $224 \times 224$ | 21M | 4.0G | 80.0 |
| RegNetY-8G [34] | $224 \times 224$ | 39M | 8.0G | 81.7 |
| RegNetY-16G [34] | $224 \times 224$ | 84M | 16.0G | 82.9 |
| ViT-B/16 [13] | $224 \times 224$ | 86M | 55.4G | 77.9 |
| ViT-L/16 [13] | $224 \times 224$ | 307M | 190.7G | 76.5 |
| DeiT-T [42] | $224 \times 224$ | 5M | 1.3G | 72.2 |
| DeiT-S [42] | $224 \times 224$ | 22M | 4.6G | 79.8 |
| DeiT-B [42] | $224 \times 224$ | 86M | 17.5G | 81.8 |
| DeiT-B↑ [42] | $384 \times 384$ | 86M | 55.4G | 83.4 |
| Conformer-T [33] | $224 \times 224$ | 23.5M | 5.2G | 81.3 |
| Conformer-S [33] | $224 \times 224$ | 37.7M | 10.6G | 83.4 |
| Conformer-B [33] | $224 \times 224$ | 83.3M | 23.3G | 84.1 |
| PVT-T [47] | $224 \times 224$ | 13.2M | 1.9G | 75.1 |
| PVT-S [47] | $224 \times 224$ | 24.5M | 3.8G | 79.8 |
| PVT-M [47] | $224 \times 224$ | 44.2M | 6.7G | 81.2 |
| PVT-L [47] | $224 \times 224$ | 61.4M | 9.8G | 81.7 |
| Swin-T [27] | $224 \times 224$ | 29M | 4.5G | 81.3 |
| Swin-S [27] | $224 \times 224$ | 50M | 8.7G | 83.0 |
| Swin-B [27] | $224 \times 224$ | 88M | 15.4G | 83.5 |
| Swin-B [27] | $384 \times 384$ | 88M | 47G | **84.5** |
| HRFormer-T | $224 \times 224$ | 8.0M | 1.8G | 78.5 |
| HRFormer-S | $224 \times 224$ | 13.5M | 3.6G | 81.2 |
| HRFormer-B | $224 \times 224$ | 50.3M | 13.7G | 82.8 |

Table 7: **Study of the 3×3 depth-wise convolution in FFN.** We report the top1 acc., mIoU, and AP on ImageNet val, PASCAL-Context test, and COCO pose estimation val, respectively. Results on PASCAL-Context are evaluated with single-scale testing. The number of parameters and FLOPs are measured on ImageNet.

| Method | #param. | FLOPs | ImageNet | PASCAL-Context | COCO |
|---|---|---|---|---|---|
| FFN w/o 3×3 DW-Conv. | 7.9M | 1.76G | 77.83 | 46.84 | 66.88 |
| FFN w/ 3× 3 DW-Conv. | 8.0M | 1.83G | 78.48 | 49.74 | 70.92 |

## 4.4 Ablation Experiments

**Influence of $3 \times 3$ depth-wise convolution within FFN** We study the influence of the $3 \times 3$ depth-wise convolution within FFN based on HRFormer-T in Table 7. We observe that applying $3 \times 3$ depth-wise convolution in FFN significantly improves the performance on multiple tasks, including ImageNet classification, PASCAL-Context segmentation, and COCO pose estimation. For example, HRFormer-T + FFN w/ $3 \times 3$ depth-wise convolution outperforms HRFormer-T + FFN w/o $3 \times 3$ depth-wise convolution by $0.65\%$, $2.9\%$ and $4.04\%$ on ImageNet, PASCAL-Context and COCO, respectively.

**Influence of shifted window scheme & 3×3 depth-wise convolution within FFN based on Swin-T.** We compare our method with the shifted windows scheme of Swin transformer [27] in Table 8. For fair comparisons, we construct a Intra-Window transformer architecture following the same

Table 8: Influence of shifted window scheme & 3×3 depth-wise convolution within FFN based on Swin-T.

| Method | 3× 3 depth-wise convolution in FFN | #param. | FLOPs | ImageNet top1 acc. |
|--------|-----------------------------------|---------|-------|--------------------|
| Swin-T | ✗ | 28.3M | 4.5G | 81.3 |
| Swin-T | ✓ | 28.5M | 4.6G | 82.2 |
| IntraWin-T | ✗ | 28.3M | 4.5G | 80.2 |
| IntraWin-T | ✓ | 28.5M | 4.6G | 82.3 |

Table 9: Shifted window scheme v.s. 3× 3 depth-wise convolution within FFN based on HRFormer-T.

| shifted window scheme | 3×3 depth-wise convolution within FFN | #param. | FLOPs | ImageNet top1 acc. | PASCAL-Context mIoU | COCO AP |
|-----------------------|---------------------------------------|---------|-------|--------------------|--------------------|---------|
| ✗ | ✓ | 8.0M | 1.8G | 78.5 | 49.7 | 70.9 |
| ✓ | ✗ | 7.9M | 1.6G | 76.6 | 43.3 | 67.3 |

architecture configurations of Swin-T [27] except that we do not apply shifted windows scheme. We see that applying 3×3 depth-wise convolution within FFN improves both Swin-T and Intrawin-T. Surprisingly, when equipped with 3× 3 depth-wise convolution within FFN, Intrawin-T even outperforms Swin-T.

**Shifted window scheme v.s. 3×3 depth-wise convolution within FFN based on HRFormer-T.** In Table 9, we compare the 3 × 3 depth-wise convolution within FFN scheme to the shifted window scheme based on HRFormer-T. According to the results, we see that applying 3×3 depth-wise convolution within FFN significantly outperforms applying shifted window scheme across all different tasks.

**Comparison to ViT, DeiT & Swin on pose estimation.** We report the COCO pose estimation results based on the two well-known transformer models, including ViT-Large [13], DeiT-B🡅 [42] and Swin-B [27] in Table 10. Notably, both ViT-Large and Swin-B[‡] are pre-trained on ImageNet21K in advance and then finetuned on ImageNet1K and achieve 85.1% and 86.4% top-1 accuracy respectively. DeiT-B🡅 is trained on ImageNet1K for 1000 epochs and achieves 85.2% top-1 accuracy. We apply deconvolution modules to upsample the output representations of the encoder following the SimpleBaseline [49] for three methods. The number of parameters and FLOPs are listed on the fourth and fifth columns of Table 10. According to the results in Table 10, we see that our HRFormer-B achieves better performance than all three methods with fewer parameters and FLOPs.

**Comparison to HRNet.** We compare our HRFormer to the convolutional HRNet with almost the same architecture configurations via replacing all the transformer blocks with the conventional basic block consisting of two 3 × 3 convolutions. Table 11 shows the comparison results on ImageNet, PASCAL-Context, and COCO. We observe that HRFormer significantly outperforms HRNet under various configurations with much less model and computation complexity. For example, HRFormer-T outperforms HRNet-T by 2.0%, 1.5%, and 1.6% on three tasks while requiring only around 50% parameters and FLOPs, respectively. In summary, HRFormer achieves better performance via exploiting the benefits of transformers such as content-dependent dynamic interactions.

## 5 Conclusion

In this work, we present the High-Resolution Transformer (HRFormer), a simple yet effective transformer architecture, for dense prediction tasks, including pose estimation and semantic segmentation. The key insight is to integrate the HRFormer block, which combines local-window self-attention and FFN with depth-wise convolution to improve the memory and computation efficiency, with the multi-resolution parallel design of the convolutional HRNet. Besides, HRFormer also benefits from adopting convolution in the early stages and mixing short-range and long-range attention with multi-scale fusion scheme. We empirically verify the effectiveness of our HRFormer on both pose estimation and semantic segmentation tasks.

Table 10: **Comparisons to ViT & DeiT on COCO pose estimation** `val`. ‡ marks the methods pretrained on ImageNet-22K.

| Method | image size | #param. | FLOPs | COCO |
|---|---|---|---|---|
| ViT-Large‡ | $256 \times 192$ | 308.5M | 60.1G | 69.2 |
| DeiT-B🐟 | $256 \times 192$ | 90.0M | 17.9G | 69.0 |
| Swin-B‡ | $256 \times 192$ | 93.2M | 17.6G | 74.3 |
| HRFormer-B | $256 \times 192$ | 43.2M | 12.2G | 75.6 |

Table 11: **Comparisons to HRNet.** We report the top1 acc., mIoU, and AP on ImageNet `val`, PASCAL-Context `test`, and COCO pose estimation `val`, respectively. Results on PASCAL-Context are based on single-scale testing. The number of parameters and FLOPs are measured on ImageNet.

| Method | #param. | FLOPs | ImageNet | PASCAL-Context | COCO |
|---|---|---|---|---|---|
| HRNet-T | 15.6M | 2.7G | 76.5 | 47.8 | 69.3 |
| HRFormer-T | 8.0M | 1.8G | 78.5 | 49.3 | 70.9 |
| HRNet-S | 24.5M | 5.0G | 78.7 | 52.3 | 73.1 |
| HRFormer-S | 13.5M | 3.6G | 81.2 | 53.8 | 74.0 |
| HRNet-B | 85.3M | 20.3G | 81.4 | 55.2 | 75.1 |
| HRFormer-B | 50.3M | 13.7G | 82.8 | 58.5 | 75.6 |

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
