# Supplementary of "HRT: High-Resolution Transformer for Dense Prediction"

## 1  Comparison with the SOTA on semantic segmentation task.

We add the comparison with the co-current SOTA methods such as Swin [2] and DPT-Hybrid [3] on more datasets. Above all, we show that increasing the window size of the local-window attention within HRT-B from $7 \times 7$ to $15 \times 15$ gains $0.6\%$, $1.2\%$, $0.8\%$, and $2.4\%$ on Cityscapes `val`, PASCAL-Context `test`, COCO-Stuff `test`, and ADE20K `val` with slightly more parameters and FLOPs. The reason for using the large window size is that the depth of our HRT-B is relatively small. For example, HRT-B consists of only 10 transformer encoder layers (on the deepest network branch) while both Swin-S and Swin-B [2] consist of 24 transformer encoder layers.

Compared to the co-current SOTA transformer methods, HRT-B + OCR ($15 \times 15$) performs better on both Cityscapes and COCO-Stuff. For PASCAL-Context, the DPT-Hybrid [3] achieves the best performance via pre-training their models on the ADE20K. For ADE20K, HRT-B + OCR ($15 \times 15$) outperforms Swin-B + UpperNet by $0.3\%$ with $50\%$ fewer parameters, and SETR-MLA achieves the best performance on ADE20K with nearly $2\times$ more FLOPs and $5\times$ more parameters.

Table 1: **Comparison with the recent SOTA on semantic segmentation tasks.** We report the mIoUs on Cityscapes `val`, PASCAL-Context `test`, COCO-Stuff `test`, and ADE20K `val`. The number of parameters and FLOPs are measured on the image size of $1024 \times 1024$, and the output label map size of $19 \times 1024 \times 1024$. All results are evaluated with multi-scale testing. ‡: the results are obtained with extra pre-training on ADE20K. $7 \times 7$ and $15 \times 15$ marks the window size.

| Method | #params. | FLOPs | Cityscapes | PASCAL-Context | COCO-Stuff | ADE20K |
|---|---|---|---|---|---|---|
| *Transformer as backbone* | | | | | | |
| SETR-PUP [6] | 317.8M | 2326.7G | 82.2 | 55.3 | – | 50.1 |
| SETR-MLA [6] | 309.5M | 2138.6G | – | 55.8 | – | 50.3 |
| Swin-S + UperNet [2] | 81.16M | 1036.50G | – | – | – | 49.5 |
| Swin-B + UperNet [2] | 121.18M | 1187.90G | – | – | – | 49.7 |
| *CNN as backbone* | | | | | | |
| Deeplabv3 [1] | 87.1M | 1394.0G | 80.7 | 54.1 | – | – |
| PSPNet [5] | 68.0M | 1028.8G | 80.0 | 54.0 | 43.3 | – |
| HRNet-W48 + OCR [4] | 74.5M | 924.7G | – | 56.2 | 40.5 | 45.7 |
| *CNN+Transformer as backbone* | | | | | | |
| DPT-Hybrid [3] | 124.0M | 1231.5G | – | 60.5‡ | – | 49.0 |
| HRT-B + OCR ($7 \times 7$) | 56.0M | 1051.6G | 82.0 | 57.3 | 42.5 | 47.6 |
| HRT-B + OCR ($15 \times 15$) | 56.2M | 1119.9G | 82.6 | 58.5 | 43.3 | 50.0 |

Submitted to 35th Conference on Neural Information Processing Systems (NeurIPS 2021). Do not distribute.

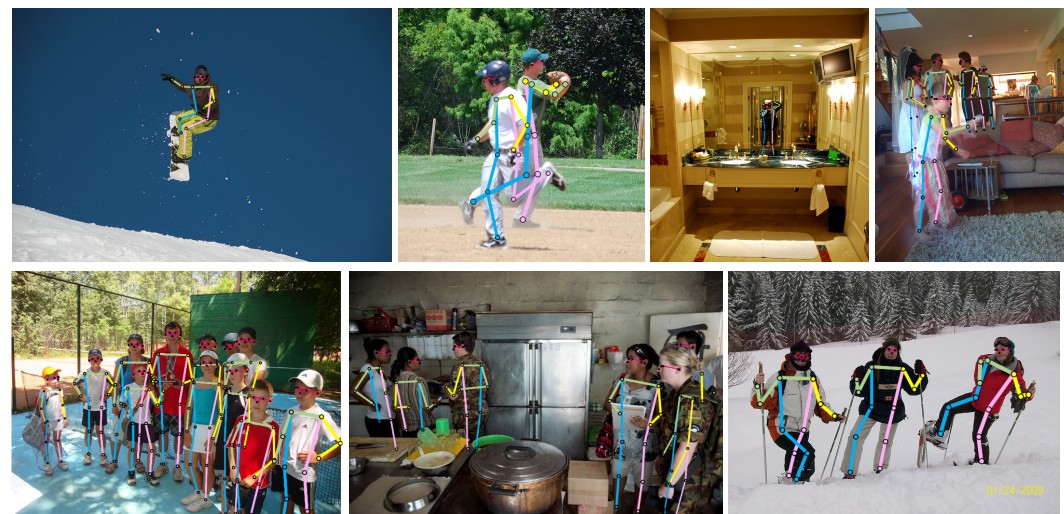

Figure 1: Visualization of the pose estimation results based on HRT-B on COCO `val`.

## 2 More Visualization Results.

We present additional visualizations of the example results of our method on both pose estimation and semantic segmentation tasks. Figure 1 shows more pose estimation results of HRT-B on COCO `val`. Figure 2 shows more semantic segmentation results on Cityscapes `val`, PASCAL-Context `test` and COCO-Stuff `test`.

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

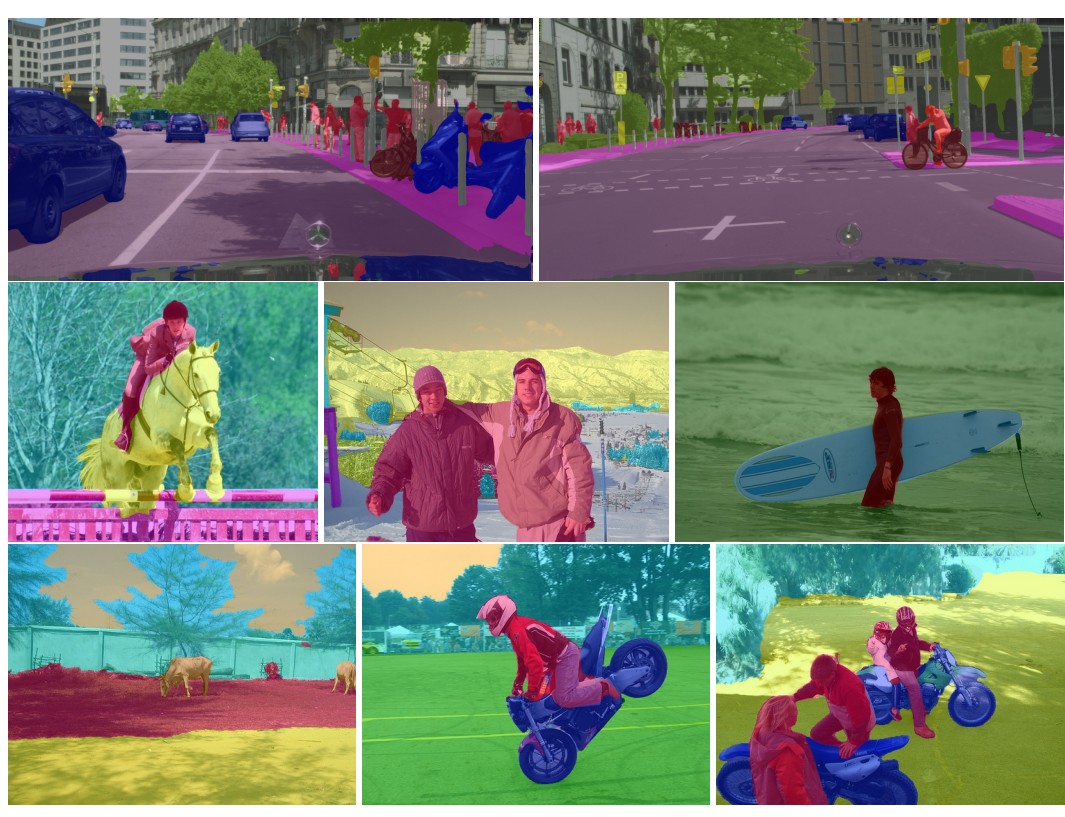

Figure 2: Visualization of the semantic segmentation results based on HRT-B + OCR on Cityscapes `val`, PASCAL-Context `test`, and COCO-Stuff `test`.