# OpenReview forum: "HRFormer: High-Resolution Vision Transformer for Dense Predict"
_NeurIPS.cc/2021/Conference — NeurIPS 2021 Poster_

### Official Review · Reviewer_143m · 2021-07-10

**Rating:** 6
**Confidence:** 4

**Summary:**

This paper proposes a way to replace the main convolution layers in the HRNet with self-attention layers. A feedforward network with 3x3 depth-wise convolution is also proposed to replace the MLP in transformer. The proposed High-Resolution Transformer achieves performance improvements for dense prediction tasks such as pose estimation and semantic segmentation while saving memory and computation.

**Limitations And Societal Impact:**

There is no description of limitation in this paper.

**Main Review:**

The method presented in this paper is not very novel, but technically sound.
Strong points:
- The proposed FFN completes the task of information exchange between different windows well.
- Experimental results on pose estimation and semantic segmentation show that the proposed High-Resolution Transformer is better than HRNet in terms of performance, memory cost and computation cost.
Weak points:
- Not very novel. The main innovation of this paper is to design a FFN with 3x3 depth-wise convolution to perform information exchange across different windows. The rest is simply replacing the main structure of HRNet with the local attention layer.
- There is no description of limitation in this paper. Considering that DETR has the disadvantage of slow convergence, I wonder how the convergence speed of the architecture proposed in this paper compares with HRNet?


**Time Spent Reviewing:**

3

---

> ### Author Response · Authors · 2021-08-09
> **Response to Reviewer 143m**
>
>
> We thank the reviewer for the careful reviews and constructive suggestions. We answer the questions as follows.
>
>
> ___
> >"**Not very novel. The main innovation of this paper is to design a FFN with 3x3 depth-wise convolution to perform information exchange across different windows. The rest is simply replacing the main structure of HRNet with the local attention layer.**"
>
> A: Thanks for your suggestions. The main innovation of our work is to introduce a new HRT architecture that suits various dense prediction tasks and the HRT is constructed with an efficient HRT transformer block that combines the local-window self-attention and FFN with 3x3 depth-wise convolution. We summarize the comparisons with various vision transformer on dense prediction tasks in the following Tables.
>
> - **Advantages of HRT on pose estimation task.**
>
> We train all methods with exactly the same settings as our HRT-B for fairness. We can see that our HRT-B achieves better COCO pose estimation results with fewer parameters and FLOPs. More detailed results are shown in Table 9.
>
> | Method      | input size | #params | FLOPs | COCO AP |
> | ----------- | ----------- | ----------- | ----------- | -----------:|
> | ViT-Large (pretrained on ImageNet-22K)  | 256×192 | 308.5M | 60.1G | 69.2 |
> | DeiT-B (pretrained on ImageNet-1K)   | 256×192 | 88.9M | 67.9G | 71.8 |
> | Swin-B (pretrained on ImageNet-1K)  | 256×192 | 93.2M | 17.6G | 73.9 |
> | Swin-B (pretrained on ImageNet-22K) | 256×192 | 93.2M | 17.6G | 74.3 |
> | HRT-B (pretrained on ImageNet-1K)   | 256×192 | 43.2M | 12.2G | 75.6 |
>
> - **Advantages of HRT on semantic segmentation task.**
>
> We can see that our HRT-B + OCR performs best on PASCAL-Context and achieves comparable performance with SETR-PUP and SETR-MLA with much fewer parameters and FLOPs. More detailed results are summarized in Table 1 of the supplementary material.
>
> | Method      | #params | FLOPs  | ADE20K mIoU | PASCAL-Context mIoU |
> | ----------- | ----------- | -----------  | -----------:|-----------:|
> | Swin-B + UperNet[26]  | 121.2M | 1187.9G  | 49.7 | - |
> | SETR-PUP[58] | 317.8M | 2326.7G | 50.1 | 55.3 |
> | SETR-MLA[58] | 309.5M | 2138.6G | 50.3 | 55.8 |
> | HRT-B + OCR           | 56.2M  | 1119.9G  | 50.0 | 58.5 |
>
> ___
> >"**There is no description of limitation in this paper. Considering that DETR has the disadvantage of slow convergence, I wonder how the convergence speed of the architecture proposed in this paper compares with HRNet?**"
>
> A: Thanks for your suggestions. Our HRT converges as fast as the original HRNet in our experiments when choosing the AdamW solver for most experiments. We compare the intermediate ImageNet classification results based on HRNet-B and HRT-B in the following table. We would like to include the convergence curves of both HRT and HRNet in the revised version following your suggestions.
>
> | Method      | #params | FLOPs |  50-epoch | 100-epoch | 150-epoch  | 200-epoch | 250-epoch | 300-epoch |
> | ----------- | ----------- | ----------- | ----------- | ----------- | ----------- | ----------- | -----------  | -----------|
> | HRNet-B     | 85.3M | 20.3G |  72.9 |  75.4  | 77.3 |  79.1 | 80.9 | 81.4 |
> | HRT-B       | 50.3M | 13.7G |  75.4 |  77.7  | 79.6 |  81.3 | 82.5 | 82.8 |
>
> According to the above results, we can see that HRT does not suffer from a slow convergence issue. Besides, we also have compared HRT with HRNet under the same training settings (such as the same training iterations) across classification, segmentation, and pose estimation tasks in Table 9, thus, we also do not observe slow convergence issues for both segmentation and pose estimation tasks.

---

> > ### Comment · Reviewer_143m · 2021-08-15
> > **Reply: Response**
> >
> > Thank you for your reply.
> >
> > The proposed method has achieved surprising performance. Considering that it doesn't converge slowly, I believe that if there is no new work to integrate HRNet with Transformer, the proposed architecture will become the main stream backbone of future pose estimation or segmentation methods. This work is definitely above the acceptance threshold.
> >
> > However, I still think the novelty of the proposed method is limited to some extent. HRNet is still the mainly used backbone in pose estimation currently, and Swin Transformer demonstrates the capability of Transformer on computer vision. It is a natural idea to combine HRNet with Transformer (and its performance is expected to be well), especially in the context of many works emerged to integrate Transformer and CNN. While the combination of HRNet and Transformer in this work is straight forward and relatively simple.
> >
> > In conclusion, I agree that the proposed method is good, but I still think it is not very innovative. I think I'm not going to change the review.

---

> > > ### Author Response · Authors · 2021-08-29
> > > **Thanks for the response of reviewer 143m**
> > >
> > > Thanks for your encouraging response and insightful comments. We also hope our High-Resolution Transformer (HRT) architecture can become the mainstream backbone of future pose estimation or segmentation methods.

---

### Official Review · Reviewer_tHJt · 2021-07-14

**Rating:** 7
**Confidence:** 3

**Summary:**

In this paper, the authors proposed a new model framework (HRT) using transformers for dense prediction in computer vision tasks.

Comparing to conventional patch-based vision transformers for dense prediction (such as segmentation and human pose estimation), HRT has three key components: 1) pixel-based local-windowed multiheaded attention with reduced compute overhead; 2) Depth-wise convolution (inside FFN modules) to aggregate info across local-windowed attention 3) Multi-resolution framework to enhance the representation quality for dense predictions (instead of using deconv after patch-based  VIT transformer encoders).

Empirically on image classification, human pose estimation and segmentation tasks, the authors shows that HRT can attain higher accuracy at similar or lower param count / FLOPs than 1) conv / deconv baselines and 2) conventional patch-based ViT encoder + deconv baselines.

**Ethical Concerns:**

I do not see anything improper or having ethical issues with this paper.

**Limitations And Societal Impact:**

I think the attached checklist answers the questions adequately. The only thing that worth mentioning is there is no technical limitation discussion (the author give N/A for the questions but I do not see this being a factor that downgrades the score for the review.) I suggest some technical discussion on the situations where the proposed methods might not be preferred etc.

**Main Review:**

Originality: I think the techniques (including pixel-based attention, depth-with convolution in FFN and Multi-resolution backbone) are novel to be used in transformer based models for dense prediction in vision tasks. I don't think the fact that depth-width conv and multi-resolution backbone have been used in the convolutional models affects the originality of this paper. By leveraging these techniques in the paper for transformer models, it leads to a new design paradigm for attention based models for dense predictions.

Quality: Given this is empirical-study-oriented work, I think the full model accuracy demonstration is impressive across tasks using various baselines but 1) the ablation study needs to be more systematic to fully discussing the merits of the techniques and 2) Additional results/info needs to be provided to achieve a full evaluation and reproducibility. Specifically:

1. Ablation: In order to ablate out the contribution of each of the proposed technique, it should demonstrate results when each of the technique is ablated out or added (in a technique-individual fashion or an incrementally building-up fashion). In the current ablation study, when comparing to patch-based VIT, it is unclear whether the major benefits is from the pixel-based-attention or from the multi-resolution design. These information are relevant to readers who are looking for the most critical technique in a trade-off between system and statistical performance.

2. Additional results: I suggest reporting of results over multiple random seed with both averaged accuracy and error bar. This will help fully evaluate whether the results are statistically meaningful or reliable. Besides this, key hyperparameters / tuning protocol needs to be revealed to get full reproducibility.

I am willing to raise the score if the above experiment concerns are resolved.


Clarity: I think the paper is meaningfully well written. Here are a few minor suggestions on further improvement:

1. Figure 1,2 and their captions does not very explicitly cover the fact that it is pixel based attention. This potentially triggers some confusion when people step in with the typical feeling that ViT is patch-based.

2. It might be helpful to visualize the accuracy as a function of param count or flops to compare the different models / methods; this could help better demonstrate the advantage across the spectrum. The current results are organized in tables which is okey but less straight-forward to fully extract the information.

Significance: Using transformer / attention based model is an active topic for computer vision, the techniques proposed in this paper (such as pixel based attention instead of patch-based) could create very positive impact on the model design trend for various type of vision tasks.

**Time Spent Reviewing:**

6 hours

---

> ### Author Response · Authors · 2021-08-09
> **Response to Reviewer tHJt**
>
> We thank the reviewer for the careful reviews and constructive suggestions. We answer the questions as follows.
>
> ___
> >"**Comparing to conventional patch-based vision transformers for dense prediction (such as segmentation and human pose estimation), HRT has three key components: 1) pixel-based local-windowed multiheaded attention with reduced compute overhead; 2) Depth-wise convolution (inside FFN modules) to aggregate info across local-windowed attention 3) Multi-resolution framework to enhance the representation quality for dense predictions (instead of using deconv after patch-based VIT transformer encoders).**"
>
> A: Thanks for your summarization. We are not sure whether we correctly understand the meaning of "pixel-based" in your first mentioned key component “pixel-based local-windowed multiheaded attention with reduced compute overhead”. We clarify that "pixel-based" is the same as "patch-based" essentially. We guess that the caption “models the interactions between pixels within the same window” in Figure 1 might be misleading. We use the “pixels” to represent different positions in a feature map and the “pixels” on feature maps of different resolutions correspond to different patch sizes. For example, each “pixel” in 4x, 8x, 16x, and 32 resolution feature maps of HRT corresponds to patch size of 4x4, 8x8, 16x16, and 32x32 respectively. In summary, the use of "pixel-based" might be an inaccurate statement and we will improve the captions of Figure 1 to avoid confusion in the revised version.
>
> ___
> >"**Originality: I think the techniques (including pixel-based attention, depth-with convolution in FFN and Multi-resolution backbone) are novel to be used in transformer based models for dense prediction in vision tasks. I don't think the fact that depth-width conv and multi-resolution backbone have been used in the convolutional models affects the originality of this paper. By leveraging these techniques in the paper for transformer models, it leads to a new design paradigm for attention based models for dense predictions.**"
>
> A: Thanks for the insightful and positive comments. We also hope our HRT can provide “a new design paradigm for attention-based models for dense predictions”.
>
> ___
> >"**Ablation: In order to ablate out the contribution of each of the proposed techniques, it should demonstrate results when each of the techniques is ablated out or added (in a technique-individual fashion or an incrementally building-up fashion). In the current ablation study, when comparing to patch-based VIT, it is unclear whether the major benefits are from the pixel-based attention or from the multi-resolution design. These information is relevant to readers who are looking for the most critical technique in a trade-off between system and statistical performance.**"
>
> A: We follow your suggestions to study the benefits of each technique in an incrementally building-up fashion and report the results in the following Table. As illustrated in the above response, we have clarified that "pixel-based attention" is essentially the "patch-based attention" with smaller patch sizes. According to these results, we can see that (1) the major performance improvements come from applying "patch-based attention" with smaller patch sizes, and (2) the major efficiency improvements come from applying "local-Window patch-based attention + FFN w/ 3x3 conv".
>
> - Patch-based ViT-Tiny with a smaller patch size (16x16->8x8) significantly increases the performance (62.16->67.46) but suffers from heavy computation cost (1.3GFLOPs->7.2GFLOPs) (row1 v.s. row2).
> - applying local-window patch-based attention + FFN w/ 3x3 Conv achieves comparable performance (67.46->67.37) with much less FLOPs (7.2GFLOPs->4.5GFLOPs) (row2 v.s. row3).
> - applying the multi-resolution scheme significantly increases the performance (67.37->69.69) with slightly more FLOPs (4.5G->4.7G) and adding more different resolutions could further improve the performance (row3 v.s. row4).
>
> | Method      | patch size | #params | FLOPs | ImageNet top1 acc. |
> | ----------- | --------------- | ----------- | ----------- | -------------:|
> | Patch-based ViT-Tiny  | 16x16 | 5.7M | 1.3G | 62.16 |
> | Patch-based ViT-Tiny  | 8x8 | 5.7M | 7.2G | 67.46 |
> | + Local-Window Patch-based Attention + FFN w/ 3x3 Conv  | 8x8 | 5.8M | 4.5G | 67.37 |
> | + Local-Window Patch-based Attention + FFN w/ 3x3 Conv + Multi-Resolution  | 8x8,16x16 | 9.7M | 4.7G | 69.69 |
>
> Due to the heavy computation overhead of attention over input with patch size 4x4, we choose patch size 8x8 as our baseline to conduct the above ablation experiments. For multi-resolution design, we only include the results based on an architecture with two different resolutions with patch sizes 8x8 and 16x16 for convenience. We train all methods on ImageNet for 100 epochs for convenience and we would like to include these results in the revised version.
>
>
> ___
> >"**Additional results: I suggest reporting of results over multiple random seed with both averaged accuracy and error bar. This will help fully evaluate whether the results are statistically meaningful or reliable. Besides this, key hyperparameters/tuning protocol needs to be revealed to get full reproducibility.**"
>
>
> A: Good suggestions. We follow your suggestions to run our HRT-B + OCR on Cityscapes, PASCAL-Context, and COCO-Stuff multiple times over multiple random seeds and summarize the (single-scale) results in the following table.
>
> | Dataset  | Run1 | Run2 | Run3 | Mean±Standard Deviation. |
> | ----------- | ----------- | ----------- | ----------- | -----------:|
> | Cityscapes  | 81.9 | 81.7 | 82.0 | 81.8±0.082 |
> | PASCAL-Context  | 57.5 | 57.6 | 57.5 | 57.5±0.047 |
> | COCO-Stuff  | 42.1 | 42.6 | 42.4 | 42.4±0.205 |
>
> We also would like to report the performance mean and variance on both ImageNet and COCO pose estimation tasks.
> Besides, we will include a detailed table to show the key hyperparameters/tuning protocol to ensure reproducibility.
>
>
> ___
> >"**Figure 1,2 and their captions do not very explicitly cover the fact that it is pixel-based attention. This potentially triggers some confusion when people step in with the typical feeling that ViT is patch-based.**"
>
> A: Thanks for your suggestions. We have clarified the details on the "pixel-based attention". We will improve the captions of Figure 1 to avoid confusion in the revised version.
>
> ___
> >"**It might be helpful to visualize the accuracy as a function of param count or flops to compare the different models/methods; this could help better demonstrate the advantage across the spectrum. The current results are organized in tables which is okay but less straightforward to fully extract the information.**"
>
> A: Good suggestions and we will “visualize the accuracy as a function of param count or flops to compare the different models/methods to better demonstrate the advantage of the proposed method”.
>
>
> ___
> >"**I suggest some technical discussion on the situations where the proposed methods might not be preferred etc.**"
>
> A: Good suggestions. One of the main limitations of HRT is related to the running speed when compared to CNN models as Vision Transformer operations are not well supported on the current hardware.

---

> > ### Comment · Reviewer_tHJt · 2021-08-22
> > **Thanks for the efforts and clarification**
> >
> > Thanks for the clarification and additional experiments.
> >
> > I think the new ablation study reveals the contribution of each techniques. And the multi-seed results shows the results are statistically meaningful.
> >
> > Given the additional results on showing the advantage over Swin Transformers (higher accuracy at lower FLOPs or parameter counts), I believe the insights in this paper can give useful information to researchers in the similar domain.
> >
> > I would raise my score from 6 yo 7.

---

> > > ### Author Response · Authors · 2021-08-29
> > > **Thanks for the response of reviewer tHJt**
> > >
> > > Thanks for your encouraging response and for increasing the rating score. We also hope our High-Resolution Transformer (HRT) architecture can bring useful information to researchers in a similar domain.

---

### Official Review · Reviewer_G26c · 2021-07-15

**Rating:** 5
**Confidence:** 5

**Summary:**

The paper proposes a vision transformer like HRNet (a CNN-based visual architecture) for dense prediction. The architecture is tested on classification/segmentation and pose estimation.

**Limitations And Societal Impact:**

I do not see any potential negative societal impact of their work

**Main Review:**

Totally, transferring an idea (HRNet) from CNN to vision transformer is ok, but the authors should provide the rationality or effectiveness. Though the architecture is different from some recent vision transformer, the improvement is marginal.

Originality: The idea of this paper is highly like HRNet, and the attention mechanism in Figure 1 is not a new thing. Actually, most new vision transformers reduce the cost by divide-and-conquer. For example, SwinTransformer etc. all resort to self-attention in a local region.

Clarity: The paper is easy to follow.

Others (to be improved):
1. The authors cite some papers: PVT/Swin/CvT etc. in related works. So, I think the authors have read these papers before finishing their own work. Since there have been very few published vision transformers, I think it is neccessary to compare with these arxiv-ones. I think the design of PVT/Swin/CvT are simpler than this paper and they show better performance, which makes me doubt the effectiveness of HRNet in vision  transformers.

2. I think the results in Table 4 show the significance of vision transformer over CNN, but does not prove that HRT is a good choice for vision transformer.

3. A little advice: In table 3, all models take a 224x224 image as input, it is also ok to explain it in the caption rather than repeat it again and again in the table.



**Time Spent Reviewing:**

2

---

> ### Author Response · Authors · 2021-08-06
> **Response to Reviewer G26c**
>
> We thank the reviewer for the careful reviews and constructive suggestions. We answer the questions as follows.
>
> ___
> >"**Totally, transferring an idea (HRNet) from CNN to vision transformer is ok, but the authors should provide the rationality or effectiveness. Though the architecture is different from some recent vision transformer, the improvement is marginal.**"
>
> A: Thanks for your suggestions. The proposed method is mainly purposed for dense prediction tasks and we have shown the effectiveness of HRT for both pose estimation and semantic segmentation tasks. We disagree that the improvement is marginal and we have included the comparisons with the recent Swin on semantic segmentation tasks in the supplementary material to show that our HRT-B outperforms Swin-B on ADE20K while saving more than 50% parameters. We also show that HRT-B significantly outperforms various vision transformer methods such as Swin and PVT on dense prediction tasks in the following response.
>
> ___
> >"**Originality: The idea of this paper is highly like HRNet, and the attention mechanism in Figure 1 is not a new thing. Actually, most new vision transformers reduce the cost by divide and conquer. For example, SwinTransformer etc. all resort to self-attention in a local region.**"
>
> A: First, our HRT unifies the advantages of both HRNet and the self-attention mechanism. Second, we agree that the attention mechanism in Figure 1 is adopted in many recent vision transformer methods. How to connect the non-overlapping windows is the key challenge and the main difference between these methods is also how to connect the non-overlapping windows. For example, the HaloNet[42] connects different windows via expanding the key-value window sizes to gather information outside the local window, SwinTransformer[26] exploits shifted windows scheme to ensure cross-window connections, and ISANet[20] introduces a complementary partition to connect the local windows. Different from them, we apply the FFN w/ 3 × 3 convolution to ensure the cross-window connections. In summary, the combination of local-window self-attention and FFN w/ 3x3 convolution is effective for various dense prediction tasks.
>
>
> ___
> >"**The authors cite some papers: PVT/Swin/CvT etc. in related works. So, I think the authors have read these papers before finishing their own work. Since there have been very few published vision transformers, I think it is neccessary to compare with these arxiv-ones. I think the design of PVT/Swin/CvT are simpler than this paper and they show better performance, which makes me doubt the effectiveness of HRNet in vision transformers.**"
>
> A: Thanks for your suggestions. We have included the comparison to Swin on semantic segmentation in Table-1 of supplementary material. We summarize more detailed comparison results in the following Tables. According to these results, we can see that HRT performs better on both semantic segmentation and pose estimation tasks.
>
> - **Comparison with PVT[45], Swin[26] and CvT[46] on ImageNet classification val set.**
>
> According to the results, we see that HRT-B achieves comparable performance with CvT-21[46] and Swin-B[26]. Considering the HRT is purposed for dense prediction tasks, we show the advantage of HRT-B on both semantic segmentation and pose estimation tasks in the following response.
>
> | Method      | #params | FLOPs  | ImageNet top1 acc. |
> | ----------- | ----------- | -----------  | -----------:|
> | PVT-S[45]   | 25M | 3.8G  | 79.8 |
> | PVT-M[45]   | 44M | 6.7G  | 81.2 |
> | PVT-L[45]   | 61M | 9.8G  | 81.7 |
> | Swin-T[26]  | 29M | 4.5G  | 81.3 |
> | Swin-S[26]  | 50M | 8.7G  | 83.0 |
> | Swin-B[26]  | 88M | 15.4G  | 83.5 |
> | CvT-13[46]  | 20M | 4.5G  | 81.6 |
> | CvT-21[46]  | 32M | 7.1G  | 82.5 |
> | HRT-T       | 8M    | 1.8G  | 78.5 |
> | HRT-S       | 13.5M | 3.6G | 81.2 |
> | HRT-B       | 50.3M | 13.7G | 82.8 |
>
> - **Comparison with PVT[45] and Swin[26] on ADE20K segmentation val set.**
>
> The result of PVT-Large + Semantic FPN[45] is based on https://github.com/whai362/PVT/tree/v2/segmentation#results-and-models and the result of Swin-B + UperNet[26] is based on https://github.com/SwinTransformer/Swin-Transformer-Semantic-Segmentation#ade20k. According to these results, we can see that HRT-B performs better than both PVT and Swin on ADE20K segmentation with much fewer parameters and FLOPs.
>
> | Method      | #params | FLOPs  | ADE20K mIoU |
> | ----------- | ----------- | -----------  | -----------:|
> | PVT-Large + Semantic FPN[45] | 65.1M | - | 43.5 |
> | Swin-B + UperNet[26]  | 121.2M | 1187.9G  | 49.7 |
> | HRT-B + OCR           | 56.2M  | 1119.9G  | 50.0 |
>
>
> - **Comparison with Swin[26] on COCO pose estimation val set.**
>
> We train the Swin-B with exactly the same settings as our HRT-B for fairness. We can see that our HRT-B achieves better COCO pose estimation results with much fewer parameters and FLOPs.
>
> | Method      | input size | #params | FLOPs | COCO AP |
> | ----------- | ----------- | ----------- | ----------- | -----------:|
> | Swin-B (pretrained on ImageNet-1K)  | 256×192 | 93.2M | 17.6G | 73.9 |
> | Swin-B (pretrained on ImageNet-22K) | 256×192 | 93.2M | 17.6G | 74.3 |
> | HRT-B (pretrained on ImageNet-1K)   | 256×192 | 43.2M | 12.2G | 75.6 |
>
>
> ___
> >"**I think the results in Table 4 show the significance of vision transformer over CNN, but does not prove that HRT is a good choice for vision transformer.**"
>
> A: We agree that Table 4 mainly shows the significance of vision transformer over CNN. We show that HRT is a good choice for vision transformer in Table 9. In Table 4, most of the previous methods are based on CNN and there exist few vision transformer architectures that report performance on COCO pose estimation tasks. In Table 9, we report the pose estimation performance based on two vision transformer methods including ViT-Large and Deit-B. According to the results, our HRT-B outperforms both ViT-Large and Deit-B while requiring fewer parameters and FLOPs. We also report the pose estimation results with Swin-B in the following table. In summary, our HRT performs better than these strong vision transformer models on the pose estimation tasks.
>
> - **Comparison with Swin[26], ViT-Large[12], and DeiT-B[40] on COCO pose estimation val set.**
>
> | Method      | input size | #params | FLOPs | COCO AP |
> | ----------- | ----------- | ----------- | ----------- | -----------:|
> | ViT-Large (pretrained on ImageNet-22K)  | 256×192 | 308.5M | 60.1G | 69.2 |
> | DeiT-B (pretrained on ImageNet-1K)   | 256×192 | 88.9M | 67.9G | 71.8 |
> | Swin-B (pretrained on ImageNet-1K)  | 256×192 | 93.2M | 17.6G | 73.9 |
> | Swin-B (pretrained on ImageNet-22K) | 256×192 | 93.2M | 17.6G | 74.3 |
> | HRT-B (pretrained on ImageNet-1K)   | 256×192 | 43.2M | 12.2G | 75.6 |
>
>
> ___
> >"**A little advice: In table 3, all models take a 224x224 image as input, it is also ok to explain it in the caption rather than repeat it again and again in the table.**"
>
> A: Good suggestion and we will explain the input size in the caption.

---

> > ### Comment · Reviewer_G26c · 2021-08-28
> > **Thanks for the authors' careful reply**
> >
> > I have read the responses from authors and reconsider my opinions.
> >
> > The authors resolve some of my concerns, so I think I could change my rating from 4 to 5. However, I am still inclined to reject this paper for the following reasons:
> >
> > Lack of Creativity. Specifically, I think the core designs of this paper include:
> >
> > 1 Split all input tokens into different regions to save cost. As I said in my initial comments, this is not a new thing, and almost all vision transformers apply this policy (e.g., Swin, CvT, CAT, and etc.).
> >
> > 2 Replace MLP in transformers with convolutions. It is an interesting design which can bring improvements.
> >
> > 3 High Resolution designs, which is same as HRNet in CNN.
> >
> > Unfair Comparison for Some Experiments. For semantic segmention, HRViT uses transformer-based OCR as the segmentation head while other methods use CNN-based segmentation heads (e.g., Swin uses UPerNet). I think, for fair comparison, the authors should use the same segmentation head for all backbones. On ADE20K dataset, HRViT+OCR outperforms 0.3% in mIOU than Swin+UPerNet. Considering OCR is transformer-based and is newer than CNN-based UPerNet, the enhancement brings by the backbone (i.e., HRViT) is marginal.
> >
> > Marginal Improvment. HRViT achieves 81.2% accuracy in ImageNet with 3.6G FLOPs, while others like Swin achieves 81.3% with 4.5G FLOPs. Further, HRViT may feel powerless when enlarging the model (82.8% with a 13.7G-FLOPs model while Swin achieves 83.0% with much less cost (8.7G) FLOPs).
> >
> > Totally, I think the paper does not contain enough contributions that enlighten others, and its improvement is trivial.

---

> > > ### Author Response · Authors · 2021-08-29
> > > **Thanks for the response of reviewer G26c**
> > >
> > >
> > > We thank the reviewer for the careful response and for increasing the rating. We answer the questions as follows.
> > >
> > > ___
> > > >"**Lack of Creativity. Specifically, I think the core designs of this paper include: 1 Split all input tokens into different regions to save cost. As I said in my initial comments, this is not a new thing, and almost all vision transformers apply this policy (e.g., Swin, CvT, CAT, and etc.). 2 Replace MLP in transformers with convolutions. It is an interesting design which can bring improvements. 3 High Resolution designs, which is same as HRNet in CNN.**"
> > >
> > > A: Thanks for your comments. The main contribution of this work is at introducing the High-Resolution Transformer (HRT) architecture for dense prediction tasks. The design of HRT architecture combines the mentioned three factors reasonably and achieves relatively significant improvements on various dense prediction tasks.
> > >
> > > ___
> > > >"**Unfair Comparison for Some Experiments. For semantic segmention, HRViT uses transformer-based OCR as the segmentation head while other methods use CNN-based segmentation heads (e.g., Swin uses UPerNet). I think, for fair comparison, the authors should use the same segmentation head for all backbones. On ADE20K dataset, HRViT+OCR outperforms 0.3% in mIOU than Swin+UPerNet. Considering OCR is transformer-based and is newer than CNN-based UPerNet, the enhancement brings by the backbone (i.e., HRViT) is marginal.**"
> > >
> > > A: To ensure the semantic segmentation comparison experiments fair to some extent, we follow your suggestions to combine Swin-B and the transformer-based segmentation head OCR but find that Swin-B + OCR achieves slightly worse performance than the original Swin-B + UperNet on ADE20K. Therefore, HRT-B consistently outperforms Swin-B with either CNN-based segmentation head or transformer-based segmentation head. Considering that HRT-B + OCR saves more than 50% parameters, the improvement of HRT-B + OCR over Swin-B + UperNet is not marginal indeed. Besides, we also have shown that HRT-B (pre-trained on ImageNet-1K) significantly outperforms Swin-B (pre-trained on ImageNet-22K) on COCO pose estimation task. We would like to include these results in the revised version and conduct more ablations if you have any further suggestions.
> > >
> > >
> > > ___
> > > >"**Marginal Improvment. HRViT achieves 81.2% accuracy in ImageNet with 3.6G FLOPs, while others like Swin achieves 81.3% with 4.5G FLOPs. Further, HRViT may feel powerless when enlarging the model (82.8% with a 13.7G-FLOPs model while Swin achieves 83.0% with much less cost (8.7G) FLOPs). Totally, I think the paper does not contain enough contributions that enlighten others, and its improvement is trivial.**"
> > >
> > > A: The improvements in both semantic segmentation and pose estimation are relatively significant. Our High-Resolution Transformer is purposed for dense prediction tasks instead of ImageNet classification tasks. We already have shown the advantages of the High-Resolution Transformer over Swin on both semantic segmentation task and pose estimation task. In summary, our High-Resolution Transformer is a more competitive transformer architecture for dense prediction tasks.

---

### Official Review · Reviewer_Jvmd · 2021-07-16

**Rating:** 6
**Confidence:** 4

**Summary:**

This paper modifies HRNet replacing standard convolutional layers with a transformer block which consists of a local multi-head attention operation (on non-overlapping windows) followed by a residual block with a 3x3 depth-wise convolution. The proposed model is benchmarked on ImageNet, COCO pose estimation, and a number of segmentation datasets.

**Limitations And Societal Impact:**

currently no discussion is included in the submission

**Main Review:**

Strengths:
- combining the advances in dense architecture design from HRNet with transformer blocks is a good step to take, taking advantage of approaches that work well for ConvNets makes a lot of sense as we see this wave of new transformer-based models.
- thorough experiments show that the proposed block is a good upgrade to HRNet, this is especially clear from the comparison made in Table 10
- experiments across many benchmarks including classification, pose, and segmentation are comprehensive showing the proposed model is competitive with recent work
- the proposed block is clear and straightforward

Weaknesses:
- it seems that a number of relevant results are left out of the benchmark comparisons - notably results that outperform the proposed method. For example, Swin [26] is absent from the ImageNet-1K results (Table 3). I also notice that [19] and [54] are cited but not included in the results in Table 5. While getting state-of-the-art results is nice, it is important to acknowledge recent work even if it has better performance
- use of a 3x3 depth-wise convolution is a straightforward enough change that it feels as though it gets an unnecessary amount of attention in the paper (for example, I appreciate what Figure 3 is communicating, but I am not sure it is needed). Given the simplicity of the module, it would have been nice to see ablations with other strategies to pass information across windows - like, how does 3x3 conv compare to consecutive local window attention blocks with shifted windows?

Overall I am currently borderline with the paper, I think it is a nice, simple idea that clearly improves a common architecture in the literature. That said, given the simplicity I think it would be nice to see more exploration of the design decisions of the model (e.g. different window sizes at different resolutions, alternatives to pass info across windows besides a conv layer). Also, I think it would be better to not leave out relevant recent work from the benchmark results.

**Update**

I was borderline on the paper, but the authors have put a good amount of work into their rebuttal to address my concerns. While not the most technically novel, it is well motivated and executed. Overall the results are in the ballpark of other work in this space, I personally find Table 10 the most compelling case for the paper as there is plenty of work building off of HRNet, and this proposed version improves it across all tasks while using less params and compute. I raise my score from a 5 to a 6.

**Time Spent Reviewing:**

3

---

> ### Author Response · Authors · 2021-08-06
> **Response to Reviewer Jvmd**
>
> We thank the reviewer for the careful reviews and constructive suggestions. We answer the questions as follows.
>
> ___
> >"**it seems that a number of relevant results are left out of the benchmark comparisons - notably results that outperform the proposed method. For example, Swin [26] is absent from the ImageNet-1K results (Table 3). I also notice that [19] and [54] are cited but not included in the results in Table 5. While getting state-of-the-art results is nice, it is important to acknowledge recent work even if it has better performance**"
>
> A: Thanks for your suggestions. We will follow your suggestions to include these results in both Table 3 and Table 5. We include the detailed comparisons with mentioned methods [26,19,54] in the following Tables.
>
> - **Comparison with Swin[26] on ImageNet-1K classification val set.**
>
> Compared to Swin-B, our HRT-B achieves slightly worse ImageNet classification results with fewer parameters and FLOPs.
>
> | Method      | #params | FLOPs  | ImageNet top1 acc. |
> | ----------- | ----------- | -----------  | -----------:|
> | Swin-T[26]  | 29M | 4.5G  | 81.3 |
> | Swin-S[26]  | 50M | 8.7G  | 83.0 |
> | Swin-B[26]  | 88M | 15.4G  | 83.5 |
> | HRT-T       | 8M | 1.8G  | 78.5 |
> | HRT-S       | 13.5M | 3.6G | 81.2 |
> | HRT-B       | 50.3M | 13.7G | 82.8 |
>
> The main advantage of our HRT over Swin is for dense prediction tasks and we summarize the results in the following two Tables.
>
> - **Comparison with Swin[26] on ADE20K segmentation val set.**
>
> | Method      | #params | FLOPs  | ADE20K mIoU |
> | ----------- | ----------- | -----------  | -----------:|
> | Swin-B + UperNet[26]  | 121.2M | 1187.9G  | 49.7 |
> | HRT-B + OCR           | 56.2M  | 1119.9G  | 50.0 |
>
> - **Comparison with Swin[26] on COCO pose estimation val set.**
>
> | Method      | input size | #params | FLOPs | COCO AP |
> | ----------- | ----------- | ----------- | ----------- | -----------:|
> | Swin-B (pretrained on ImageNet-1K)  | 256×192 | 93.2M | 17.6G | 73.9 |
> | Swin-B (pretrained on ImageNet-22K) | 256×192 | 93.2M | 17.6G | 74.3 |
> | HRT-B (pretrained on ImageNet-1K)   | 256×192 | 43.2M | 12.2G | 75.6 |
>
> In summary, our HRT-B achieves better COCO pose estimation results and ADE20K segmentation results than Swin-B with much fewer parameters and FLOPs. More details of the segmentation results are summarized in Table 1 of the supplementary material.
>
> - **Comparison with [19,54] on COCO pose estimation val set.**
>
> | Method      | input size | #params | FLOPs  | COCO AP |
> | ----------- | ----------- | ----------- | -----------  | -----------:|
> | HRNet-W48[44]  |  384x288 |  63.6M | 32.9G | 76.7 |
> | HRNet-W48 + UDP[19]  |  384x288 |  63.6M | 32.9G | 77.2 |
> | HRNet-W48 + DARK[54] |  384x288 |  63.6M | 32.9G | 77.2 |
> | HRT-B                |  384x288 |  43.2M | 26.8G | 77.2 |
>
> The results of HRNet-W48[44] (https://github.com/open-mmlab/mmpose/blob/master/configs/body/2d_kpt_sview_rgb_img/topdown_heatmap/coco/hrnet_coco.md), HRNet-W48 + UDP[19] (https://github.com/open-mmlab/mmpose/blob/master/configs/body/2d_kpt_sview_rgb_img/topdown_heatmap/coco/hrnet_udp_coco.md), and HRNet-W48 + DARK[54] (https://github.com/open-mmlab/mmpose/blob/master/configs/body/2d_kpt_sview_rgb_img/topdown_heatmap/coco/hrnet_dark_coco.md) are based on mmpose (https://github.com/open-mmlab/mmpose).
>
> According to the above results, both UDP[19] and DARK[54] improve the performance of HRNet-W48[44] from 76.7 to 77.2. Our HRT-B already achieves 77.2 w/o using any advanced techniques. We believe that our HRT-B could achieve better results by exploiting either UDP[19] or DARK[54]. We would like to include these results in the revised version.
>
>
>
> ___
> >"**use of a 3x3 depth-wise convolution is a straightforward enough change that it feels as though it gets an unnecessary amount of attention in the paper (for example, I appreciate what Figure 3 is communicating, but I am not sure it is needed). Given the simplicity of the module, it would have been nice to see ablations with other strategies to pass information across windows - like, how does 3x3 conv compare to consecutive local window attention blocks with shifted windows?**"
>
> A: We will follow your suggestions to improve Figure 3 or move Figure 3 to the supplementary material. We add the comparisons with “other strategies to pass information across windows” (e.g., shifted windows scheme) in the following Table. According to the results, we can see that 3x3 conv performs better than shifted window scheme on COCO pose estimation. We will add more comparison results in the revised version.
>
> | Method      | Shifted Windows | FFN w/ 3 × 3 Conv | #params | FLOPs  | COCO AP |
> | ----------- | ----------- | ----------- | ----------- | -----------  | -----------:|
> | HRT-T      | ✗       | ✗       | 7.9M | 1.76G | 66.88 |
> | HRT-T      | ✗       | ✓       | 8.0M | 1.83G | 70.92 |
> | HRT-T      | ✓       | ✗       | 7.9M | 1.76G | 67.32 |
>
> ___
> >"**I think it would be nice to see more exploration of the design decisions of the model (e.g. different window sizes at different resolutions, alternatives to pass info across windows besides a conv layer). Also, I think it would be better to not leave out relevant recent work from the benchmark results.**"
>
> A: Thanks for your suggestions. We report the results with different window sizes at different resolutions on semantic segmentation tasks and we will add more results if necessary.  We use W1, W2, W3, W4 to represent the window sizes associated with feature maps with different resolutions with stride 4,8,16,32. We choose larger window sizes for higher resolution branches, thus, we have W1 > W2 > W3 > W4. According to these results, we can see that applying larger windows improves the performance, and applying different window sizes at different resolutions makes no big difference. Besides, we have compared “alternatives to pass info across windows besides a conv layer” in the above responses. We would like to follow your suggestions to include the results of relevant recent works.
>
> | Method      | W1,W2,W3,W4 |  #params | FLOPs  | PASCAL-Context mIoU |
> | ----------- | ----------- | ----------- | ----------- | -----------:|
> | HRT-B + OCR      | 7,7,7,7           |  56.0M | 1051G | 57.3 |
> | HRT-B + OCR      | 11,11,11,11       |  56.1M | 1069G | 57.6 |
> | HRT-B + OCR      | 13,13,13,13       |  56.1M | 1083G | 58.1 |
> | HRT-B + OCR      | 15,15,15,15       |  56.2M | 1120G | 58.5 |
> | HRT-B + OCR      | 15,13,11,9        |  56.1M | 1094G | 57.9 |
> | HRT-B + OCR      | 21,17,13,9        |  56.2M | 1148G | 57.9 |
> | HRT-B + OCR      | 17,15,13,11       |  56.2M | 1113G | 58.5 |
>
> We have summarized more segmentation results of HRT + OCR with window sizes as 15,15,15,15 in Table 1 of the supplementary.

---

> ### Author Response · Authors · 2021-08-30
> **Looking forward to hearing the response from reviewer Jvmd**
>
> We thank the reviewer for the previous careful reviews and constructive suggestions.
>
> We have learned a lot from the response from all the other reviewers. We also would like to hear your further comments and suggestions.

---

### Decision · Program_Chairs · 2021-09-27

**Decision:**

Accept (Poster)

**Comment:**

The authors propose a "high-resolution Transformer" architecture for dense prediction tasks using a multiresolution hierarchy and self-attention over local windows. All reviewers agreed that the ideas were sensible and technically sound, and that the experiments demonstrated improvement. There were concerns about missing comparisons (reviewers Jvmd, G26c) and missing ablations (tHJt, Jvmd), but the authors seem to have addressed all of these in rebuttal. There were also concerns of limited novelty (reviewers 143m, G26c), but I don't think this alone should disqualify the paper. Therefore I recommend acceptance.